# Sedentary Behavior, Physical Inactivity, and the Prevalence of Hypertension, Diabetes, and Obesity During COVID-19 in Brazil

**DOI:** 10.3390/ijerph22091367

**Published:** 2025-08-30

**Authors:** Jeferson Roberto Collevatti dos Anjos, Igor Massari Correia, Chimenny Auluã Lascas Cardoso de Moraes, Jéssica Fernanda Corrêa Cordeiro, Atila Alexandre Trapé, Jorge Mota, Dalmo Roberto Lopes Machado, André Pereira dos Santos

**Affiliations:** 1School of Physical Education and Sport of Ribeirão Preto, University of São Paulo, Ribeirão Preto 14040-907, Brazil; atrape@usp.br (A.A.T.); dalmo@usp.br (D.R.L.M.); andreps@alumni.usp.br (A.P.d.S.); 2Study and Research Group in Anthropometry, Training and Sport, School of Physical Education and Sport of Ribeirão Preto, University of São Paulo, Ribeirão Preto 14040-907, Brazil; igormassari@usp.br (I.M.C.); chimenny.mor@usp.br (C.A.L.C.d.M.); jessica.cordeiro@alumni.usp.br (J.F.C.C.); 3College of Nursing of Ribeirão Preto, University of São Paulo, Ribeirão Preto 14040-902, Brazil; 4Research Center in Physical Activity, Health and Leisure (CIAFEL) and Laboratory for Integrative and Translational Research in Population Health (ITR), Faculty of Sports, University of Porto, 4200-450 Porto, Portugal; jmota@fade.up.pt; 5Human Exposome and Infectious Diseases Network (HEID), Ribeirão Preto 14040-902, Brazil; 6Department of Child, Family and Population Health Nursing, University of Washington, Seattle, WA 98105, USA

**Keywords:** physical activity, SARS-CoV-2, hypertension, obesity, diabetes mellitus

## Abstract

Objectives: To analyze, across the periods before, during, and after the implementation of Social Isolation and Distancing Measures (IMDIS): (a) changes in the prevalence of non-communicable chronic diseases (NCDs), stratified by age group in the Brazilian population; and (b) the association between physical inactivity (PI), insufficient moderate-to-vigorous physical activity (MVPA), and sedentary behavior (SB) with the occurrence of these conditions. This cross-sectional study used data from VIGITEL (Brazil’s Surveillance System of Risk and Protective Factors for Chronic Diseases by Telephone Survey), an annual population-based telephone survey conducted across the country. Data were collected in 2019, 2021, and 2023, with a total sample size of 101,226 participants. Arterial hypertension (AH) and diabetes mellitus (DM) were self-reported, and obesity (OB) was diagnosed using body mass index. PI, insufficient MVPA, and SB were identified via VIGITEL indicators. Chi-square tests assessed differences in prevalence overall and by age group. Logistic regression models estimated odds ratios (ORs) for associations between demographic variables, behavioral factors, and the studied periods. The prevalence of AH and DM was highest among individuals over 60 years, reaching 61% after IMDIS, a period when OB also peaked across all age groups. Individuals aged 30–59 and those over 60 had higher odds of AH, DM, and OB across all periods. Female participants had higher ORs for AH and DM both before and after IMDIS. PI and insufficient MVPA were associated with increased odds of AH, DM, and OB in all periods, while SB significantly elevated the OR for OB at all time points. After IMDIS, there was an increase in the prevalence of AH, DM, and OB among older adults and younger individuals. PI, insufficient MVPA, SB, and advanced age were all associated with a greater likelihood of NCDs at every stage of the study. The high post-IMDIS rates of AH, DM, and OB highlight the need for urgent public health strategies. Low-cost programs, such as live videos and online group sessions, should be included in national physical activity guidelines. These initiatives are affordable, aligned with WHO goals, and reduce PI in IMDIS scenarios. Incorporating them into Academia da Saúde and Agita Brasil strengthens NCD prevention and increases the resilience of the health system for future health crises.

## 1. Introduction

The COVID-19 pandemic, declared in March 2020, resulted in a high number of deaths and increased social vulnerability worldwide [1,2]. In Brazil, by 31 December 2023, more than 38 million cases of infection had been recorded, along with 708 thousand deaths from COVID-19 in the country [3,4].

Due to the high lethality of the virus and the absence of specific medications or vaccines to combat it and its health risks, national health authorities implemented Social Distancing and Isolation Measures (IMDIS), including restrictions on gatherings, mandatory mask use, and the closure of schools, universities, gyms, and community spaces for physical activity (PA) [5,6,7,8,9].

The national literature highlights that the IMDIS implemented during the pandemic, combined with health insecurity, uncertainty, and fear of COVID-19, had significant impacts and brought major changes to the daily lives of Brazilians, especially regarding health status, quality of life, and physical activity (PA) habits. Silva et al. [10] identified a 26% increase in the prevalence of physical inactivity (PI) and a 266% increase in sedentary behavior (SB) among 39,693 adults. Togni et al. [11] showed that 55.8% of individuals who were physically active before the pandemic presented insufficient Moderate to Vigorous Physical Activity (MVPA) less than 150 min per week during the implementation of the measures. Additionally, Malta et al. [12] reported a worsening of noncommunicable chronic diseases (NCDs) during the restriction period, with increased prevalence of arterial hypertension (AH), diabetes mellitus (DM), and obesity (OB).

Given the evidence of increased prevalence of NCDs, PI, SB, and insufficient MVPA during IMDIS [10,11,12], it becomes essential to investigate these indicators in Brazil. Even before the health crisis caused by the pandemic, the high prevalence of these conditions already represented a serious public health concern [13]. Between 2013 and 2019, for example, national data showed an increase in the prevalence of AH (from 21.5% to 23.8%), DM (from 6.3% to 8.2%), and OB [14]. Moreover, 47% of the Brazilian population still does not meet the minimum PA recommendations established by the World Health Organization [15].

However, when analyzing the impacts of IMDIS on the prevalence of NCDs and PA levels, it is essential to consider sociodemographic disparities, especially those related to age groups, as they may have influenced how the population responded to the restrictions imposed by IMDIS [16]. International studies show that NCDs are unevenly distributed by age: the prevalence of AH and DM increases markedly after the age of 45, while OB also rises among young adults (20–40 years) [17]. This dynamic leads to more significant harm among age groups considered more vulnerable, such as older adults and middle-aged individuals. Furthermore, PA levels varied significantly across different age groups during the COVID-19 IMDIS, highlighting the importance of strategies tailored to specific age profiles [18,19].

In this context, the Surveillance System for Risk and Protective Factors for Chronic Diseases by Telephone Survey (VIGITEL) stands out as an essential tool for investigating the effects of IMDIS on NCDs, PI, SB, and insufficient MVPA before, during, and after the pandemic. In continuous operation since 2006, VIGITEL provides systematic, representative, and low-cost data, supporting the development of public health promotion policies tailored to the Brazilian context [20].

Although the effects of IMDIS on NCDs, PI, SB, and insufficient MVPA are well established in the literature [11,12,13], analyses in Brazil that structure these indicators by age group in an integrated way are still lacking. This gap is particularly relevant because international studies show distinct patterns of NCDs according to age [20]. Furthermore, age groups showed different changes in PA levels during IMDIS [18,19]. Thus, the present study aimed to analyze, in the periods before, during, and after the IMDIS: (a) changes in the prevalence of NCDs, stratified by age group in the Brazilian population; and (b) the association between PI, insufficient MVPA, and SB with the occurrence of these conditions.

## 2. Materials and Methods

### 2.1. Type of Study and Context

This is a cross-sectional observational epidemiological study based on data from VIGITEL. Implemented in 2006, VIGITEL is an annual national telephone survey conducted in all 26 Brazilian state capitals and the Federal District. It aims to monitor the prevalence and distribution of the main risk and protective factors related to NCDs among the adult and older population (≥18 years) [21].

### 2.2. Study Population and Sampling

VIGITEL data collected between 2019, 2021, and 2023 were used, with a total sample size of 101,226 participants. This temporal choice is justifiable because it coincides with the periods before, during, and after the IMDIS in Brazil. Until 2021, VIGITEL collected probability samples of the adult population living in households with at least one landline, totaling approximately 2000 individuals in the capitals and the Federal District. However, between 2020 and 2021, the sample was reduced to approximately 1000 individuals per city due to limitations imposed by the COVID-19 pandemic [22].

In 2023, a minimum of 800 interviews were established per location, grouped into 400 by landline and 400 by cell phone. This reduction occurred as a result of delays in the bidding process for data collection in 2022, which was carried out at the end of that year, and in the first half of 2023. Estimates on the frequency of risk and protective factors were calculated with a 95% confidence interval, presenting a maximum error of four percentage points for the total adult population and five percentage points for sex-specific estimates, assuming equivalent proportions of men and women in the sample [23].

The VIGITEL sampling process is conducted in two stages. Initially, a random sample of telephone lines is extracted from the database of the National Telecommunications Agency (Anatel). These lines are then randomly distributed into subsamples for each city. In the second stage, an adult resident in the selected household, or the adult user of the mobile phone number, is invited to participate in the interview [21].

VIGITEL estimates were adjusted using a post-stratification weight, designed to correct for the unequal probability of selection, especially in households with more than one adult or a fixed telephone line. In addition, this adjustment aims to align the distribution of the interviewed sample (by sex, age, and education) with the projected distribution of the total population in each city and year. This process is performed using the Rake method [24], using Census data and official population projections.

Detailed information on the sampling process, weighting, and the evolution of VIGITEL over the years is available in the VIGITEL databases, which are accessible to the public (https://svs.aids.gov.br/download/Vigitel/), accessed date at 11 May 2024. The VIGITEL data collection was approved by the National Committee for Ethics in Research involving Human Beings of the Ministry of Health, under number 65610017.1.0000.0008.

### 2.3. Data Collection and Organization

The VIGITEL data collection instrument was a questionnaire that addressed questions about sociodemographic characteristics; lifestyle; self-reported weight and height, self-assessment of health status and previous medical diagnosis of AH, DM, and depression; performance of exams for early detection of cancer in women; health insurance and issues related to traffic situations. In the present study, questions related to sociodemographic characteristics (Age Group: 18–29 years; 30–59 years, and ≥60 years; Sex: Male, Female), PA practice, and NCDs were used [21,22,23].

The data of participants aged ≥18 years who responded to the survey in the years 2019 (*n* = 52,443), 2021 (*n* = 27,093), and 2023 (*n* = 21,690) were combined into a single database in Excel, which allowed for analyses and comparisons between the different moments of the IMDIS. Individuals aged >18 years who had complete information on self-reported AH, self-reported DM, OB, and who answered the complete VIGITEL PA questionnaire were included.

### 2.4. Self-Reported Arterial Hypertension

The presence of AH was determined by the self-reported diagnosis of AH, assessed by the question: “Has a doctor ever told you that you have AH?” The following were considered to have a diagnosis of AH: adults who answered yes, in addition to verifying, in the case of women, those who answered “no” to the question about gestational AH (Did the AH occur only during some period of pregnancy?) [23].

### 2.5. Self-Reported Diabetes Mellitus

The presence of DM was determined by the self-reported diagnosis of the disease, assessed by the question: “Has a doctor ever told you that you have diabetes?” The following were considered to have a diagnosis of diabetes: adults who answered yes, in addition to verifying, in the case of women, those who answered “no” to the question about gestational diabetes (Did this condition occur only during some period of pregnancy?) [23].

### 2.6. Obesity

Anthropometric measurements of weight (kg) and height (meters) were obtained by self-report from the following questions: “Do you know your weight (even if it is an approximate value)?” and “Do you know your height?”. The participant’s Body Mass Index (BMI) was calculated by dividing body weight in kilograms by height in meters squared (kg/m^2^) [25]. Obesity was considered a BMI > 30 kg/m^2^ for adults and the elderly [26], using the hot deck technique to impute missing weight and height data from VIGITEL [23].

### 2.7. Insufficient Physical Activity Variables

The variables of PI, SB, and insufficient MVPA were developed based on the questions in the PA practice block of the VIGITEL questionnaire. This block addresses the weekly frequency of PA and the duration of activities. It also explores the types of activities performed, such as walking, running, cycling, and sports, and considers the practice of PA in contexts such as leisure, commuting, work, and household chores. Finally, it assesses SB, considering the daily time spent on activities such as watching TV or using electronic devices per day [21].

Based on this information, the system performs routines to analyze PA measurements in the different domains (leisure, commuting, household, and work) and generates indicators that classify individuals into PA categories that include sufficient and insufficient MVPA, physically active and inactive, as well as SB and non-sedentary. These classifications are presented in the survey spreadsheet in a dichotomous manner. PI was defined as the absence of PA practice in the last 12 months [22]. Insufficient MVPA was classified as individuals who did not achieve >150 min of moderate PA or >75 min of vigorous PA or an equivalent combination of these two intensities per week, considering the domains of leisure time, commuting, and occupational [27]. SB was classified according to total screen time greater than 3 h/day considering cell phone, computer, tablet, and television [23].

### 2.8. Statistical Analysis

The data were extracted from the public VIGITEL databases. The information related to the years 2019 (before), 2021 (during), and 2023 (after) was consolidated into a single database in a Microsoft Excel^®^ spreadsheet, which allowed comparisons between the years. The coding was then performed independently by two researchers, allowing the coding to be validated at a later stage by double entry.

Absolute frequencies were generated for AH, DM, OB, PI, insufficient MVPA, and SB. The analyses were performed for the total population and stratified by age groups in different IMDIS periods. Pearson’s chi-square test was performed to assess the association between the prevalence of AH, DM, OB, PI, insufficient MVPA, and SB and the different IMDIS moments. Logistic regression models were applied to estimate the Odds Ratio (OR) in relation to the presence of AH, DM, and OB, taking into account variables such as sex, age group, PI, SB, and insufficient MVPA. For data analysis, the IBM SPSS Statistics 23.0 software was used, adopting a confidence level of α = 5% in all analyses.

## 3. Results

Figure 1 presents four charts that illustrate the distribution of the total study population (N = 101,226) during the periods before, during, and after the IMDIS in terms of sex and age group. There is a predominance of women in all categories, with percentages of 64.7% in the total population (Figure 1a), 65% before the IMDIS (Figure 1b), 65.8% during the IMDIS (Figure 1c), and 62.5% after the IMDIS (Figure 1d). The age groups show the highest frequency of individuals in the 30 to 59 age range, as evidenced in Figure 1a (44.4%) and Figure 1d (52.5%). On the other hand, Figure 1b (44.5%) and Figure 1c (48.6%) indicate higher frequencies of individuals ≥ 60 years old.

Figure 2 shows the prevalence of AH, DM, and OB at different times during the IMDIS (Figure 2a). When analyzing the descriptive data, significant associations were observed between the highest percentages of AH and DM and the period during the IMDIS (*p* < 0.05). In addition, a trend of continuous increase in the prevalence of OB was observed, with an association between the highest percentages of OB and the period after the IMDIS. The analysis by age group revealed significant associations between AH in individuals aged 18 to 29 years and ≥60 years, demonstrating that the highest percentages of AH are associated with the period after the IMDIS for these age groups (Figure 2b). Among individuals aged 18 to 29 years and those aged 30 to 59 years, an association was observed between the highest frequencies of DM and the period during the implementation of the measures. For individuals ≥ 60 years, the highest frequencies of DM were significantly associated with the period after IMDIS (Figure 2c). OB showed significant associations in the age groups of 18 to 29 years, 30 to 59 years, and ≥60 years, showing a continuous increasing trend, with the highest levels of OB associated with the period after IMDIS (Figure 2d).

Figure 3 shows the variation in PI, SB, and insufficient MVPA, total and stratified by age group, before, during, and after the IMDIS in adults and older adults in Brazil. The statistical analysis revealed significant variations in the number of individuals with these behaviors throughout the different moments of the IMDIS. The highest frequencies of PI and insufficient MVPA were associated with the period during the IMDIS, while the highest percentages of SB were associated after this period (Figure 3a). For the age group of 18 to 29 years, the lowest frequencies of PI were associated with the period during the IMDIS, while the highest levels of this behavior were observed after the implementation of the measures. In contrast, among individuals aged 30 to 59 years and those over 60 years, the highest PI records were associated with the period during the IMDIS (Figure 3b). Regarding SB, significant changes were noted for individuals aged 30 to 59 years and ≥60 years. The highest frequencies of SB were associated with the period during the IMDIS for those aged 30–59 years, while for individuals aged ≥60 years, the associations were between periods during and after the IMDIS, with the highest percentages of SB (Figure 3c). Variations in the levels of insufficient MVPA were observed in the age groups of 18–29 years, 30–59 years, and >60 years, with the highest percentages of this behavior associated with the period during the IMDIS (Figure 3d).

Table 1 shows the OR for the occurrence of AH before, during, and after the IMDIS, considering variables of sex, age group, insufficient MVPA, PI, and SB. The odds of AH were high among individuals aged 30 to 59 years and those ≥60 years, as well as in females. In addition, an increase in the odds of AH was observed among individuals with insufficient MVPA and PI. Regarding SB, an inverse relationship was identified, where people with higher SB had lower chances of AH during IMDIS 0.94 (95% CI: 0.89 to 0.99; *p* = 0.028).

In Table 2, it can be observed that the odds of DM were significantly higher among individuals in the age groups of 30 to 59 years and among those ≥60 years. In addition, a protective effect of female sex on the occurrence of DM was observed before IMDIS. However, after IMDIS, women showed an increased odds of DM. Insufficient levels of MVPA and PI were also associated with an increased risk of DM at all time points analyzed.

Table 3 presents the odds ratios for OB and variables of sex, age group, and PI. At all-time points of the study, individuals aged 30 to 59 years had higher odds of OB, while those aged ≥60 years had higher odds before and during IMDIS. The odds of OB were high among individuals with insufficient MVPA, PI, and SB (at the three time points analyzed).

## 4. Discussion

Data from 101,226 participants of VIGITEL were used to analyze, in the periods before, during, and after IMDIS, the variations in the prevalence of NCDs by age group in the Brazilian population, as well as the associations between PI, SB, and insufficient MVPA with the occurrence of these conditions. As far as is known, this is the first study to stratify DCNTs and PA indicators by age group during the IMDIS phases in Brazil. Although estimates of the prevalence of DCNTs, PI, SB, and insufficient MVPA have already been reported in previous studies [12], no integrated age-group analysis had been conducted until now. This study provides updated contributions to understanding the health status of the Brazilian population in the context of IMDIS. It highlights the most vulnerable age groups during the implementation of these measures, serving as important input for the formulation of public health policies aimed at promoting the population’s quality of life and well-being, while also complementing the body of evidence already established in the scientific literature.

Our results show an increase in the prevalence of AH and DM in the extreme age groups (18–29 and >60 years) after IMDIS, whereas OB reached the highest percentages in 2023 across all age groups. Comparable studies point to similar trends. Trimarco et al. [28], who analyzed medical records of 200,000 adults over seven years, observed a continuous increase in the incidence of AH, especially among older adults. The rate rose from 2.11 cases per 100 person-years (95%CI: 2.08–2.15) between 2017–2019 and 5.20 (95%CI: 5.14–5.26) between 2020 and 2022, reaching 6.76 (95%CI: 6.64–6.88) in 2023, the year following the pandemic. Similarly, a U.S. study identified a sustained increase in OB prevalence in states that implemented more stringent social distancing measures, with higher OB levels two years after the end of the restrictions [29].

In contrast to the findings of the present study, Lee et al. [30] conducted a cross-sectional, population-based study with a robust sample of 3,208,710 Korean adults, analyzing long-term trends in the prevalence of individuals diagnosed and undergoing treatment for AH between 2009 and 2022, a period that includes the COVID-19 pandemic. Although a general increase in the prevalence of AH was observed over the 14 years, the authors identified a deceleration in diagnosis and treatment rates during IMDIS compared to the pre-pandemic period. While growth slowed during IMDIS, the prevalence of AH did not decline; among young adults (19–39 years), it remained above 25%, whereas among older adults (≥60 years), the upward trend was less pronounced.

International literature and our data support that blood pressure control was significantly impaired, especially among older adults, due to the interruption of clinical follow-up, increased SB, psychological stress, dietary changes, and difficulties in accessing medications [31]. In Bangladesh, Kanti Mistry et al., [32] reported an increase in the prevalence of AH among older adults, from 43.7% in 2020 to 56.3% in 2021, highlighting the direct impact of restrictions on the cardiovascular health of this population. Complementarily, a meta-analysis involving more than six million participants showed that, although telemedicine was a relevant strategy during the pandemic, older individuals with AH showed lower adherence to this model and faced access barriers, which contributed to an increase in adverse events and worsening of blood pressure control [33]. In Brazil, Malta et al. [12] also identified an increase in the prevalence of AH, OB, and DM during the periods in which IMDIS were in effect, corroborating the trend observed internationally and reinforcing the consistency of the findings of the present study.

The fact that OB peaked in 2023 across all age groups indicates that the effects of IMDIS are long-lasting. International studies suggest that health events can impact health patterns for up to three years after their occurrence [34]. This highlights the need for continuous monitoring and phased actions to mitigate the effects of such crises. The findings indicate that the effects of IMDIS transcended the immediate period, persisting into subsequent phases. This continuity may be related to unhealthy lifestyle factors, such as chronic stress, deterioration of mental health, and interruptions in healthcare during IMDIS [35,36].

The IMDIS period was characterized by significant changes in health-related behaviors, which varied according to age group [19]. An increase was observed in PI, SB, unhealthy dietary habits, and the consumption of alcohol and tobacco [37]. On the other hand, there was also greater preparation of home-cooked meals, promoting healthier eating and a reduction in the consumption of ultra-processed foods [38]. Home-based PA [39] and the expanded use of telemedicine [40] also stood out. These findings highlight a duality in health behaviors, reflecting different forms of adaptation to the circumstances imposed by IMDIS in various population contexts. Inequalities in economic conditions, access to healthcare services, and individual adaptive capacity may explain the observed association between AH and DM among young and older adults, as well as OB across all age groups, consistent with findings in the literature [41].

Although longitudinal evidence explaining the increase in the prevalence of DCNTs after IMDIS remains limited, a plausible justification for the results found, in light of the literature, is that many patients with DCNTs did not have adequate access to treatment during this period. There was a reduction in hospitalizations for the treatment of DCNTs, motivated by patients’ and their families’ fear of contamination by the SARS-CoV-2 virus [42].

In this context, it is believed that the adverse effects of IMDIS became more evident after their conclusion. Moreover, there is evidence that health crises can negatively impact population lifestyles for up to three years [43], which reinforces the findings of this study and may, in part, explain the increase in the prevalence of DCNTs after IMDIS. Still, divergences in study designs, sampling procedures, population characteristics, and DCNT diagnosis methodologies justify the variations observed in the results [44,45]. Therefore, the challenge of establishing clear relationships between the increase or reduction in DCNTs by age group during IMDIS remains, reinforcing the need for more studies to deepen the understanding of the impacts of these measures on population health.

The increase in the odds of OB, AH, and DM among older adults is consistent with international literature [46] and may be attributed to physiological changes related to aging, such as arterial stiffness, endothelial dysfunction, and insulin resistance [47], in addition to behavioral factors such as inadequate diet, sedentary lifestyle, and limited access to healthcare services [48].

Women showed a higher risk for AH and DM, which may be explained by hormonal and social factors that influence their PA and dietary patterns [49,50]. During IMDIS, these vulnerabilities may have been exacerbated by specific barriers, such as the accumulation of domestic responsibilities and limited access to safe spaces for PA practice [51,52].

The literature shows that insufficient MVPA is directly associated with a higher prevalence of DCNTs and is one of the main determinants of global mortality [53]. SB, in turn, is widely recognized as a risk factor for OB, as it contributes to a positive energy balance [54,55,56,57]. The lower odds ratio for AH in the pre-IMDIS period may reflect higher levels of PA, favoring blood pressure control [58,59]. Strategies to promote PA and reduce SB remain fundamental for the prevention of DCNTs [60,61].

Among young people, a reduction in PI was observed alongside an increase in AH. This finding reinforces the multifactorial nature of the disease, which involves not only PA but also inadequate diet, consumption of ultra-processed foods [62], stress, anxiety, and worsening mental health [63]. These factors may explain the increase in AH even among physically active individuals, as pointed out in the literature [46,64].

This analysis highlights the need for future research that considers multiple health determinants beyond demographic and behavioral aspects to better understand the impacts of IMDIS on DCNTs. The findings reinforce the importance of an interprofessional dialog among scientists, health professionals, and policymakers, in order to support public policies adapted to age-specific and social particularities, with a focus on mitigating the effects of IMDIS. Among the practical implications, the urgency of programs that promote regular PA and the prevention of DCNTs in different population groups stands out. These results are relevant to guide the actions of health professionals in the current epidemiological scenario and support the formulation of public policies aimed at health promotion and the reduction in inequalities intensified during the pandemic.

The cross-sectional design of the study limits causal inferences and is subject to selection bias. The underrepresentation of men may compromise the generalizability of the findings, as unbalanced samples can affect the validity of the observed associations. The predominance of women in VIGITEL, therefore, should be considered a methodological limitation. Another relevant factor is the change in sample composition: from 2019 to 2021, data were collected via landline telephone, while in 2023, mobile phones were also included, which may have introduced bias in regions with lower socioeconomic coverage. Still, VIGITEL applies post-stratification weights to adjust the data to the target population, increasing its representativeness [22]. The use of self-reported data may also underestimate the prevalence of AH and DM, especially among individuals with limited access to healthcare services. However, previous studies have demonstrated acceptable validity of these measures for population surveillance purposes [65,66].

Among the strengths, the large sample size and national coverage of VIGITEL stand out, with data collected during a historic health event. The stratification by age group allowed the identification of the most vulnerable populations to the effects of IMDIS, contributing relevant evidence to the formulation of public health strategies.

Finally, the results reinforce the need for strategies aimed at reducing DCNTs in specific age groups and point to directions for future longitudinal research. Investigations that consider variables such as income, education, race/skin color, and geographic region will be essential to deepen the understanding of the impacts of IMDIS and to support more equitable and effective interventions in the post-pandemic context.

## 5. Conclusions

The prevalence of AH and DM among young and older individuals increased after IMDIS. OB showed an increasing trend across all age groups. Insufficient MVPA, PI, and SB were associated with a higher risk of NCDs, independent of the analyzed time point, with females and older age groups presenting a higher risk at certain times. Structured actions that result in increased PI and SB must be recognized as potential health risk indicators in future public health emergencies.

## Figures and Tables

**Figure 1 ijerph-22-01367-f001:**
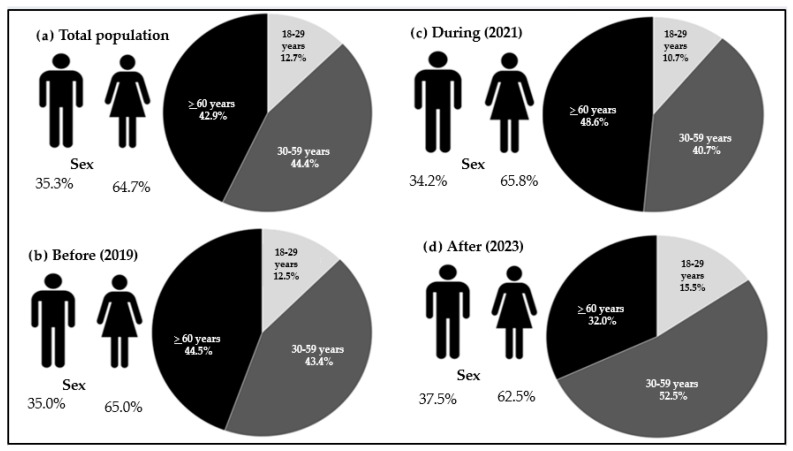
Distribution of the adult population (>18 years) according to: (**a**) sex and age group, considering the moments (**b**) before, (**c**) during, and (**d**) after the IMDIS period. Data from VIGITEL, Brazil, 2019–2023.

**Figure 2 ijerph-22-01367-f002:**
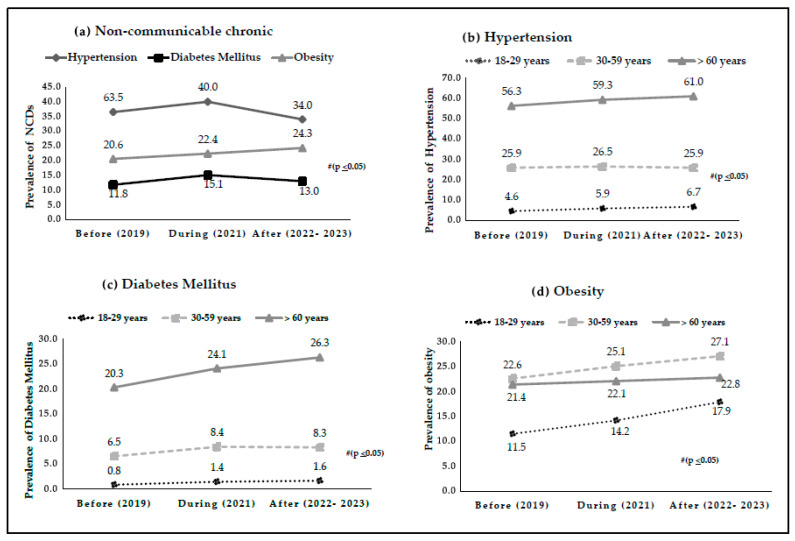
Variation in the prevalence of: (**a**) NCDs, (**b**) AH, (**c**) DM, (**d**) OB, and by age group before, during, and after the period of isolation and social distancing measures in adults and older adults in Brazil. Data from VIGITEL, Brazil, 2019–2023. # (*p* < 0.05) Pearson’s Chi-Square test.

**Figure 3 ijerph-22-01367-f003:**
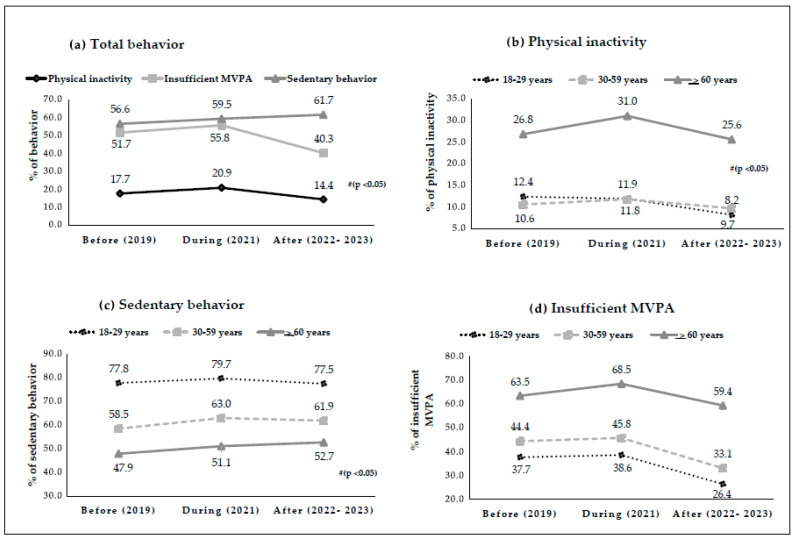
Variation in (**a**) Total Behavior, (**b**) PI (**c**) SB, and (**d**) insufficient MVPA by age group before, during, and after the period of isolation and social distancing measures in adults and older adults in Brazil. Data from VIGITEL, Brazil, 2019–2023. # (*p* < 0.05) Pearson’s Chi-Square test.

**Table 1 ijerph-22-01367-t001:** Odds ratio for the risk of Arterial hypertension and demographic variables, Physical activity practice, and Sedentary Behavior before, during, and after the IMDIS in Brazil. Data from VIGITEL, Brazil, 2019–2023.

Arterial Hypertension										
Variables		Before (2019)			During (2022)			After (2022–2023)		
		OR	IC95%	*p*-Value	OR	IC95%	*p*-Value	OR	IC95%	*p*-Value
**Age Group**										
	18–29 years	1.00 (Ref.)			1.00 (Ref.)			1.00 (Ref.)		
	30–59 years	7.05	6.25–7.94	**<0.001**	5.52	4.70–6.48	**<0.001**	4.74	4.11–5.46	**<0.001**
	≥60 years	9.02	8.65–9.64	**<0.001**	8.59	7.86–9.39	**<0.001**	8.80	8.06–9.61	**<0.001**
**Sex**										
	Male	1.00 (Ref.)			1.00 (Ref.)			1.00 (Ref.)		
	Female	1.14	1.10–1.19	**<0.001**	1.10	1.03–1.16	**0.002**	1.12	1.06–1.21	**<0.001**
**Insufficient MVPA**										
	No	1.00 (Ref.)			1.00 (Ref.)			1.00 (Ref.)		
	Yes	1.29	1.23–1.34	**<0.001**	1.35	1.27–1.44	**<0.001**	1.33	1.23–1.43	**<0.001**
**Physical Inactivity**										
	No	1.00 (Ref.)			1.00 (Ref.)			1.00 (Ref.)		
	Yes	1.24	1.17–1.31	**<0.001**	1.17	1.08–1.25	**<0.001**	1.15	1.04–1.02	**0.005**
**Sedentary Behavior**										
	No	1.00 (Ref.)			1.00 (Ref.)			1.00 (Ref.)		
	Yes	0.97	0.93–1.00	0.081	0.94	0.89–0.99	**0.028**	0.97	0.91–1.04	0.379

Captions: OR = Odds ratio; 95%CI = 95% confidence interval. *p*-value of binary logistic regression analysis. Reference of the dependent variable—Referred AH: Individuals who answered yes to the medical diagnosis of AH. Evidence of association with statistical significance is presented in bold. Source: VIGITEL database 2019–2023.

**Table 2 ijerph-22-01367-t002:** Odds ratio for the risk of Diabetes Mellitus and demographic variables, Physical activity practice, and Sedentary behavior before, during, and after the IMDIS in Brazil. Data from VIGITEL, Brazil, 2019–2023.

Diabetes Mellitus										
Variables		Before (2019)			During (2022)			After (2022–2023)		
		OR	IC95%	*p*-Value	OR	IC95%	*p*-Value	OR	IC95%	*p*-Value
** *Age Group* **										
	18–29 years	1.00 (Ref.)			1.00 (Ref.)			1.00 (Ref.)		
	30–59 years	8.34	6.35–10.96	**<0.001**	6.25	4.56–8.58	**<0.001**	5.37	4.07–7.08	**<0.001**
	≥60 years	9.54	8.28–10.99	**<0.001**	7.75	6.58–9.14	**<0.001**	7.93	6.81–9.22	**<0.001**
** *Sex* **										
	Male	1.00 (Ref.)			1.00 (Ref.)			1.00 (Ref.)		
	Female	0.92	0.86–0.97	**0.003**	1.01	0.93–1.09	0.815	1.12	1.02–1.22	**0.013**
** *Insufficient MVPA* **										
	No	1.00 (Ref.)			1.00 (Ref.)			1.00 (Ref.)		
	Yes	1.27	1.19–1.36	**<0.001**	1.25	1.15–1.36	**<0.001**	1.38	1.25–1.52	**<0.001**
** *Physical Inactivity* **										
	No	1.00 (Ref.)			1.00 (Ref.)			1.00 (Ref.)		
	Yes	1.43	1.33–1.53	**0.005**	1.35	1.23–1.47	**<0.001**	1.29	1.15–1.45	**<0.001**
** *Sedentary Behavior* **										
	No	1.00 (Ref.)			1.00 (Ref.)			1.00 (Ref.)		
	Yes	0.97	0.91–1.02	0.205	0.97	0.90–1.04	0.374	0.93	0.86–1.02	0.110

Captions: OR = Odds ratio; 95%CI = 95% confidence interval. *p*-value of binary logistic regression analysis. Reference of the dependent variable—DM reported: Individuals who answered yes to the medical diagnosis of DM. Evidence of association with statistical significance is presented in bold. Source: VIGITEL database 2019–2023.

**Table 3 ijerph-22-01367-t003:** Odds ratio for the risk of Obesity and demographic variables, Physical activity practice, and Sedentary behavior before, during, and after the IMDIS in Brazil. Data from VIGITEL, Brazil, 2019–2023.

Obesity										
Variables		Before (2019)			During (2022)			After (2022–2023)		
		OR	IC95%	*p*-Value	OR	IC95%	*p*-Value	OR	IC95%	*p*-Value
** *Age Group* **										
	18–29 years	1.00 (Ref.)			1.00 (Ref.)			1.00 (Ref.)		
	30–59 years	2.29	2.10–2.48	**<0.001**	2.02	1.81–2.27	**<0.001**	1.71	1.55–1.89	**<0.001**
	≥60 years	1.32	1.25–1.39	**<0.001**	1.10	1.02–1.18	**0.011**	0.95	0.88–1.03	0.203
** *Sex* **										
	Male	1.00 (Ref.)			1.00 (Ref.)			1.00 (Ref.)		
	Female	0.96	0.92–1.01	0.114	1.05	0.99–1.12	0.101	1.02	0.96–1.09	0.442
** *Insufficient MVPA* **										
	No	1.00 (Ref.)			1.00 (Ref.)			1.00 (Ref.)		
	Yes	1.28	1.21–1.34	**<0.001**	1.32	1.23–1.41	**<0.001**	1.28	1.19–1.39	**<0.001**
** *Physical Inactivity* **										
	No	1.00 (Ref.)			1.00 (Ref.)			1.00 (Ref.)		
	Yes	1.25	1.18–1.33	**<0.001**	1.21	1.11–1.30	**<0.001**	1.22	1.11–1.35	**<0.001**
** *Sedentary Behavior* **										
	No	1.00 (Ref.)			1.00 (Ref.)			1.00 (Ref.)		
	Yes	1.14	1.09–119	**<0.001**	1.14	1.07–1.21	**<0.001**	1.17	1.10–1.25	**<0.001**

Captions: OR = Odds ratio; 95%CI = 95% confidence interval. *p*-value of binary logistic regression analysis. Dependent variable reference—OB: Self-reported weight and height and BMI = >30.0 kg/m^2^ for adults and 27.0 kg/m^2^ for the older adults. Evidence of association with statistical significance is presented in bold. Source: VIGITEL database 2019–2023.

## Data Availability

Detailed information on the sampling process, weighting and the evolution of VIGITEL over the years is available in the VIGITEL databases, which are accessible to the public (https://svs.aids.gov.br/download/Vigitel/). Accessed on 11 May 2024.

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
