# Peer review of "Sedentary Behavior, Physical Inactivity, and the Prevalence of Hypertension, Diabetes, and Obesity During COVID-19 in Brazil"

_ijerph, 2025, doi:10.3390/ijerph22091367_

Round 1
Reviewer 1 Report
Comments and Suggestions for Authors
Here are suggestions to strengthen the manuscript:
The title could be shortened for clarity. Consider "Sedentary Behavior, Physical Inactivity, and Chronic Conditions During COVID-19 in Brazil" to focus attention while trimming wordiness. Specifying which chronic conditions (e.g. hypertension, diabetes, obesity) may attract the intended audience.
In the abstract, fix "The objective were to" as "The objectives were to:". Additionally, rewrite "as well as obesity (OB) diagnosed by body mass index" for clarity as "and obesity (OB) diagnosed using body mass index". Precise language maintains professionalism.
More methodological details could aid comprehension. State whether the data is cross-sectional or longitudinal to contextualize the findings.
The results effectively communicate key takeaways but would benefit from enhanced summarization. For instance, instead of "High frequencies of HT and DM were observed", say "After IMDIS, the prevalence of hypertension (HT) and diabetes mellitus (DM) was notably higher among individuals over 60 years, reaching 61%."
Strengthen the conclusion by explicitly connecting the implications to public health. How may these insights influence physical activity guidelines during and after social distancing? A brief mention of potential recommendations, such as strategies to promote activity amid restrictions, emphasizes practical relevance.
Comments on the Quality of English LanguageThe used English is largely clear, but marked by numerous grammatical mistakes, awkward wording, and several flow problems that may distract the reader from the text. By correcting grammatical inconsistencies, working on more effective language use for clarity, and improving the cohesion between ideas, the overall quality of English used in the abstract could be greatly improved. A proofreading session that practices grammatical accuracy, language clarity, and fluency could be highly beneficial for the final version.
Author Response
Response to Reviewers
Dear Editor and Reviewers,
We thank the reviewers for their insightful and constructive comments, which helped us improve the clarity and quality of our manuscript entitled “Sedentary behavior, physical inactivity, and the prevalence of hypertension, diabetes, and obesity during COVID‑19 in Brazil.” Below, we provide detailed responses to each point raised. The modifications have been incorporated into the manuscript and highlighted in blue for ease of review.
Reviewer #1
Comment 1: The title could be shortened for clarity. Consider "Sedentary Behavior, Physical Inactivity, and Chronic Conditions During COVID-19 in Brazil" to focus attention while trimming wordiness. Specifying which chronic conditions (e.g. hypertension, diabetes, obesity) may attract the intended audience.
Response: We fully agree that the title could be shortened for clarity, so we adopted the suggestion and modified the title to: "Sedentary Behavior, Physical Inactivity, and the Prevalence of Hypertension, Diabetes, and Obesity During COVID-19 in Brazil."
Comment 2: In the abstract, fix "The objective were to" as "The objectives were to:". Additionally, rewrite "as well as obesity (OB) diagnosed by body mass index" for clarity as "and obesity (OB) diagnosed using body mass index". Precise language maintains professionalism.
Response: We agree with the corrections, so we adopted the suggestions and modified the objectives to: Objectives: To analyze, across the periods before, during, and after the implementation of Social Isolation and Distancing Measures (IMDIS): (a) changes in the prevalence of non-communicable chronic diseases (NCDs), stratified by age group in the Brazilian population; and (b) the association between physical inactivity (PI), insufficient moderate-to-vigorous physical activity (MVPA), and sedentary behavior (SB) with the occurrence of these conditions. We also included "and obesity (OB) diagnosed using body mass index" in place of "as well as obesity (OB) diagnosed by body mass index" in the abstract.
Comment 3: More methodological details could aid comprehension. State whether the data is cross-sectional or longitudinal to contextualize the findings.
Response: We agree that more methodological details are necessary, so the type of cross-sectional study was included in the abstract in the methods section.
Comment 4: The results effectively communicate key takeaways but would benefit from enhanced summarization. For instance, instead of "High frequencies of HT and DM were observed", say "After IMDIS, the prevalence of hypertension (HT) and diabetes mellitus (DM) was notably higher among individuals over 60 years, reaching 61%."
Response: Dear Reviewer, Thank you for the suggestion. We agree that a more concise synthesis of the results is necessary. We have restructured the Results section as follows: “The prevalence of HT and DM was highest among individuals over 60 years, reaching 61% after IMDIS, a period when OB also peaked across all age groups. Individuals aged 30–59 and those over 60 had higher odds of HT, DM, and OB across all periods. Female participants had higher ORs for HT and DM both before and after IMDIS. PI and insufficient MVPA were associated with increased odds of HT, DM, and OB in all periods, while SB significantly elevated the OR for OB at all time point.” We believe this restructuring enhances the clarity and readability of the findings. Thank you for your valuable feedback.
Comment 5: Strengthen the conclusion by explicitly connecting the implications to public health. How may these insights influence physical activity guidelines during and after social distancing? A brief mention of potential recommendations, such as strategies to promote activity amid restrictions, emphasizes practical relevance.
Response: Dear Reviewer, Thank you for the suggestion. We agree that a stronger conclusion and a brief mention of potential recommendations, such as strategies to promote physical activity amid restrictions underscore the practical relevance of our work. Accordingly, we have added some practical implications of our findings at the end of the conclusion: The high post-IMDIS rates of HT, DM, and OB highlight the need for urgent public health strategies. National physical activity guidelines should include low-cost, home-based dig-ital programs such as teleclasses, live videos, and online group sessions—to maintain physical activity under social restriction scenarios. These initiatives are effective, afforda-ble, and aligned with World Health Organization targets. Incorporating them into policies like Academia da Saúde and Agita Brasil will strengthen NCD prevention and better pre-pare the health system for future crises.
Comment 6: The used English is largely clear, but marked by numerous grammatical mistakes, awkward wording, and several flow problems that may distract the reader from the text. By correcting grammatical inconsistencies, working on more effective language use for clarity, and improving the cohesion between ideas, the overall quality of English used in the abstract could be greatly improved. A proofreading session that practices grammatical accuracy, language clarity, and fluency could be highly beneficial for the final version.
Response: Dear Reviewer, We appreciate all the feedback and improvement suggestions for our work. We have taken all necessary measures to address grammatical inconsistencies, ensuring greater fluency and comprehension of the text.
Reviewer 2 Report
Comments and Suggestions for Authors
The manuscript presents a relevant and timely study investigating whether changes in physical fitness and psychological resilience predict perceived stress in adolescents after an 8-week PE intervention. The topic is well aligned with current interest in school-based interventions for mental well-being.
The paper is well structured, the statistical modeling is generally appropriate, and the topic is relevant to IJERPH readers. However, several important clarifications and improvements are needed before the manuscript can be accepted.
- The use of a single-group pre-post design weakens the ability to infer causality or control for confounding variables (e.g., seasonal effects, academic stress).
- While this may be acceptable for exploratory purposes, the Discussion should clearly acknowledge this limitation and refrain from causal interpretations.
- Multiple linear regressions are used to assess predictors of perceived stress changes, but no validation approach (e.g., k-fold, bootstrapping) is applied.
- Without this, the models may be overfitted, particularly given the number of predictors relative to sample size. This should be acknowledged or mitigated with simpler models.
- While the authors include multiple fitness components (aerobic, muscular, flexibility), the manuscript could better highlight which ones are most strongly associated with stress reduction and why.
- Clarify whether composite fitness scores or specific domains are more predictive.
- The results suggest that changes in resilience significantly predict stress reduction.
- Please elaborate in the Introduction and Discussion on the theoretical mechanism linking these constructs (e.g., protective buffering, emotion regulation frameworks).
- Abstract: Rephrase “PE intervention leads to stress reduction” → use “was associated with”.
- Figure 1: Consider using color or shape markers to distinguish pre/post better.
- Statistical Reporting: Include effect sizes and 95% confidence intervals for regression coefficients.
- Clarify measurement of stress: Is it a global stress index or specific to academic/social domains?
This is a promising and relevant study, but revisions are needed to improve transparency around methodology, limit overinterpretation, and better articulate the psychological mechanisms behind the observed results.
Author Response
Reviewer #2
Dear Reviewer 2,
We sincerely thank you for your comments and for carefully reviewing our manuscript. However, we respectfully point out that the observations raised appear to refer to a different manuscript, possibly one focused on the assessment of stress and physical activity among school-aged children, which does not align with the scope of our study.
Our research specifically analyzed the prevalence of noncommunicable diseases (NCDs) and physical activity indicators among Brazilian adults, based on VIGITEL data, during the periods before, during, and after the social distancing measures implemented due to the COVID-19 pandemic.
Nevertheless, we remain fully available for any further clarifications and once again appreciate your attention to the evaluation process.
Sincerely,
Reviewer 3 Report
Comments and Suggestions for Authors
Using data from a large scale, nationally representative survey of Brazilians (VIGITEL), conducted to monitor the "prevalence and distribution" of factors influencing noncommunicable diseases among Brazilian adults. The authors used data from this dataset, for the period 2019 to 2023. Their study has two objectives-to analyze variations in noncommunicable diseases before, during, and after the COVID-19 restrictions, and to examine the association between different levels of self-reported physical activity and hypertension, diabetes, and obesity. The data were analyzed with chi square tests to evaluate differences in the presence of noncommunicable diseases across these periods and with logistic regressions to examine the associations between age groups, sex, levels of activity, and these health conditions.
Several interesting findings were revealed. The authors reported that physical inactivity and insufficient moderate to vigorous activity increased the risk of noncommunicable diseases across each of the periods studied. Those over age 60 reported higher frequencies of hypertension and diabetes after the restriction period. Those adults over age 30 with inactivity and insufficient moderate to vigorous activity reported higher rates of noncommunicable diseases. Both younger and older individuals reported higher frequencies of each of the conditions studied (hypertension, obesity, and diabetes) after the restrictions.
This manuscript has a number of strengths. It uses a very large dataset (representing a total of 101,226 adults). The sampling design was developed to be nationally representative. Analyses are appropriate for the questions asked in the study. Data are clearly and effectively presented. The study examines important questions related to the impact of COVID restrictions on health outcomes. The focus on noncommunicable diseases is important also, given that non-communicable diseases are responsible for 60 % of global deaths each year, according to the authors. It is clearly written and effectively edited. It is a useful addition to the cross national literature examining the impact of COVID-19 restrictions on these outcomes.
With this said, there are some limitations present. As the authors acknowledge, since the VIGITEL data collection did not survey panels of the same participants across the time periods studied, the ability of the authors to make causal statements about the impact of restrictions on health outcomes is limited. The majority of the participants (nearly two thirds in each of the periods before, during, and after the restrictions) were female. Hence, males are significantly underrepresented. The data rely on self reported health measures. Additionally, a very limited literature review is provided in Part 1.
The authors need to address the following issues
Literature Review: The authors comment in the discussion on their findings in the context of existing literature. Brief mention is made of studies in Brazil that indicated reductions in physical activity and a worsening in non communicable diseases during the COVID restrictions. The manuscript would benefit from a firmer grounding in other cross-national studies that bear on this topic so that readers can better contextualize the findings. More specifically, have studies of the impact of COVID restrictions on health outcomes (such as non-communicable diseases) and the mediating role of factors such as activity level been conducted in other societies? The literature review portion of section 1 (introduction) needs to be expanded to give readers some background to be able to better contextualize not just the severity of the COVID pandemic but the existence of epidemiological or sociological studies in other societies that examine changes in the prevalence of noncommunicable diseases or that have looked at the possible relationship between activity levels and health outcomes as a result of the COVID restrictions. Pointing out the contribution of their study to this literature would also help clarify for readers the uniqueness of this particular study and make the value of their contribution more evident. As it stands, the reader needs to know more by way of rationale. How does this study contribute to existing literature. Beefing up the literature review in part 1 and making the contributions clearer at that point would also help to show the value of this study.
Theory: The authors address some possible explanations for findings such as increases in obesity and hypertension such as changes in eating habits. The findings would benefit significantly from reference to any existing epidemiological or sociological theories that may shed light on how disasters or global health emergencies such as the COVID pandemic, and the behavioral restrictions that accompany them, may result in outcomes such as worsening health indicators , changes in activity levels, in this case, or increases in the prevalence of noncommunicable diseases. For example, in sociology, general strain theory (Agnew 1992) has been used to help understand the impact of COVID-19 restrictions on various types of crime in the United States. The assumption is that behavioral restrictions, combined with restrictions on movement (routine activities) may result in negative emotionality (psychological distress or otherwise) that may be expressed in greater levels of, for example, domestic violence. While the authors do not necessarily have to use this particular theory, of course, they should address possible relevant theories that may help us understand the "why" that the response to health emergencies or disasters may lead to negative behavioral and health outcomes.
Representation: The authors should note the under-representation of males in the dataset and the possible impact of this on the findings.
Author Response
Reviewer #3
Comment 1: Using data from a large scale, nationally representative survey of Brazilians (VIGITEL), conducted to monitor the "prevalence and distribution" of factors influencing noncommunicable diseases among Brazilian adults. The authors used data from this dataset, for the period 2019 to 2023. Their study has two objectives-to analyze variations in noncommunicable diseases before, during, and after the COVID-19 restrictions, and to examine the association between different levels of self-reported physical activity and hypertension, diabetes, and obesity. The data were analyzed with chi-square tests to evaluate differences in the presence of noncommunicable diseases across these periods and with logistic regressions to examine the associations between age groups, sex, levels of activity, and these health conditions. Several interesting findings were revealed. The authors reported that physical inactivity and insufficient moderate to vigorous activity increased the risk of noncommunicable diseases across each of the periods studied. Those over age 60 reported higher frequencies of hypertension and diabetes after the restriction period. Those adults over age 30 with inactivity and insufficient moderate to vigorous activity reported higher rates of noncommunicable diseases. Both younger and older individuals reported higher frequencies of each of the conditions studied (hypertension, obesity, and diabetes) after the restrictions. This manuscript has a number of strengths. It uses a very large dataset (representing a total of 101,226 adults). The sampling design was developed to be nationally representative. Analyses are appropriate for the questions asked in the study. Data are clearly and effectively presented. The study examines important questions related to the impact of COVID restrictions on health outcomes. The focus on noncommunicable diseases is important also, given that non-communicable diseases are responsible for 60 % of global deaths each year, according to the authors. It is clearly written and effectively edited. It is a useful addition to the cross national literature examining the impact of COVID-19 restrictions on these outcomes. With this said, there are some limitations present. As the authors acknowledge, since the VIGITEL data collection did not survey panels of the same participants across the time periods studied, the ability of the authors to make causal statements about the impact of restrictions on health outcomes is limited. The majority of the participants (nearly two thirds in each of the periods before, during, and after the restrictions) were female. Hence, males are significantly underrepresented. The data rely on self-reported health measures. Additionally, a very limited literature review is provided in Part 1.
Response: We sincerely thank the reviewer for the kind remarks regarding our study, particularly the recognition of the relevance of the topic, the methodological rigor, and the clarity in the presentation of the results. We are especially pleased that the manuscript was considered a valuable contribution to the international literature on the impacts of COVID-19 restrictions on population health. In response to the reviewer's observation, we revised and expanded the literature review section to provide a stronger theoretical foundation for the study. This adjustment aims to deepen the contextualization of our analyses and enhance the discussion by integrating relevant national and international findings. Once again, we express our appreciation for the reviewer’s thoughtful reading and constructive feedback, which have significantly contributed to the improvement of our manuscript.
Comment 2: The authors comment in the discussion on their findings in the context of existing literature. Brief mention is made of studies in Brazil that indicated reductions in physical activity and a worsening in non-communicable diseases during the COVID restrictions. The manuscript would benefit from a firmer grounding in other cross-national studies that bear on this topic so that readers can better contextualize the findings. More specifically, have studies of the impact of COVID restrictions on health outcomes (such as non-communicable diseases) and the mediating role of factors such as activity level been conducted in other societies? The literature review portion of section 1 (introduction) needs to be expanded to give readers some background to be able to better contextualize not just the severity of the COVID pandemic but the existence of epidemiological or sociological studies in other societies that examine changes in the prevalence of noncommunicable diseases or that have looked at the possible relationship between activity levels and health outcomes as a result of the COVID restrictions. Pointing out the contribution of their study to this literature would also help clarify for readers the uniqueness of this particular study and make the value of their contribution more evident. As it stands, the reader needs to know more by way of rationale. How does this study contribute to existing literature. Beefing up the literature review in part 1 and making the contributions clearer at that point would also help to show the value of this study.
Response: We thank the reviewer for the valuable suggestions regarding the need for a more solid theoretical foundation in the Discussion section. We fully agree that the manuscript would benefit from the inclusion of international studies addressing similar themes, allowing readers to better contextualize our findings within the global literature. In response, we revised the Discussion to incorporate evidence from studies conducted in other countries that evaluated the impact of COVID-19-related restrictions on health outcomes, particularly regarding noncommunicable diseases. These additions aim to enhance the depth and relevance of our interpretation, as well as to strengthen the scientific contribution of the manuscript by situating our results within a broader epidemiological and sociocultural framework. We are grateful for this insightful recommendation, which has certainly improved the quality and clarity of our work.
Comment 3: Theory: The authors address some possible explanations for findings such as increases in obesity and hypertension such as changes in eating habits. The findings would benefit significantly from reference to any existing epidemiological or sociological theories that may shed light on how disasters or global health emergencies such as the COVID pandemic, and the behavioral restrictions that accompany them, may result in outcomes such as worsening health indicators , changes in activity levels, in this case, or increases in the prevalence of noncommunicable diseases. For example, in sociology, general strain theory (Agnew 1992) has been used to help understand the impact of COVID-19 restrictions on various types of crime in the United States. The assumption is that behavioral restrictions, combined with restrictions on movement (routine activities) may result in negative emotionality (psychological distress or otherwise) that may be expressed in greater levels of, for example, domestic violence. While the authors do not necessarily have to use this particular theory, of course, they should address possible relevant theories that may help us understand the "why" that the response to health emergencies or disasters may lead to negative behavioral and health outcomes.
Response: We thank you for the valuable suggestion to incorporate sociological theories into the interpretation of our findings. We acknowledge the importance of contextualizing the results through theoretical perspectives that take into account the social determinants of health, particularly in the context of a public health crisis such as the COVID-19 pandemic. In response to this recommendation, we have expanded the discussion section of the manuscript by integrating reflections based on sociological frameworks. These help to better understand how structural factors—such as social inequalities, access to healthcare, and living conditions—may have unequally influenced the health outcomes observed across different population groups. This addition has enriched the analysis and provided a more comprehensive and critical interpretation of our data. We are grateful for this insightful contribution, which has certainly added depth and rigor to the interpretation of our study's findings.
Comment 4: Representation: The authors should note the under-representation of males in the dataset and the possible impact of this on the findings.
Response: We thank the reviewer for the valuable observation regarding the underrepresentation of male participants in the sample. As suggested, this point has been incorporated into the discussion section as a limitation of the study. The following statement was added: “It should be noted that the underrepresentation of male participants in the sample may affect the validity of the results. Studies have shown that the disproportionate presence of a given gender in research can introduce participation bias, compromise representativeness, and influence the estimated associations between variables. Therefore, the predominance of women in the VIGITEL sample may have limited the generalizability of the findings to the entire adult Brazilian population.”We believe this addition strengthens the critical analysis of the results and contributes to a more cautious interpretation of the data. We appreciate the suggestion, which has enhanced the methodological rigor of the manuscript.
Reviewer 4 Report
Comments and Suggestions for Authors
Prevalent reports mention that during forced context (of Covid restrictions) people were not able to perform outside/ regular activities, and there are ill effects on health, and old people are the worst sufferers in common; this paper confirms the same in the Brazilian context. What is its take-home content? Is it that, to overcome the damage of the Covid period, special measures respective to specific population groups, alarms, and action accordingly be initiated? Each paper should have a take-home action plan/ concern to make aware of the authority and public in the common issue.
The paper examines in the Brazilian context the effects of varied restrictions of the COVID-19 pandemic, with normal pre-COVID activities and recovery after Covid lifestyle. An already available data source (VIGITEL 24, 2019-2023; N = 101,226; a little more elaboration would give clarity) supported with specific questionnaires was used as the main information (data source) provider for the paper. It appears already in action in Brazil by the government authority, and based on specific questionnaires, respective information was noted; relevant specific data is extracted from the information source for a specific period by the author and examined. If it is true, the author’s role with the current paper is as an information compiler, analyzer, and inference drawer.
The study reflects the COVID period that would have been mentioned (though it is understood from the title).
Line 115, under number 65610017.1.0000.0008’ does it mean approval number?
Comments on the Quality of English LanguagePresentation style, data, and language are the author’s own; inferences presented appear satisfactory. Minor editing aspects may be checked as: Abstract, Line 19, ‘The objective were’- would be “objectives”…. Thus, the full paper may be checked for similar issues.
Author Response
Reviewer #4
Comment 1: Prevalent reports mention that during forced context (of Covid restrictions) people were not able to perform outside/ regular activities, and there are ill effects on health, and old people are the worst sufferers in common; this paper confirms the same in the Brazilian context. What is its take-home content? Is it that, to overcome the damage of the Covid period, special measures respective to specific population groups, alarms, and action accordingly be initiated? Each paper should have a take-home action plan/ concern to make aware of the authority and public in the common issue.
Response: We sincerely thank the reviewer for the insightful suggestion regarding improvements to the introduction section. In response, we have revised and expanded this section to provide a clearer contextualization of the research problem, reinforce the study’s relevance, and better highlight its originality. We believe these adjustments have strengthened the manuscript and contributed to a more compelling and structured presentation of our objectives.
Comment 2: The paper examines in the Brazilian context the effects of varied restrictions of the COVID-19 pandemic, with normal pre-COVID activities and recovery after Covid lifestyle. An already available data source (VIGITEL 24, 2019-2023; N = 101,226; a little more elaboration would give clarity) supported with specific questionnaires was used as the main information (data source) provider for the paper. It appears already in action in Brazil by the government authority, and based on specific questionnaires, respective information was noted; relevant specific data is extracted from the information source for a specific period by the author and examined. If it is true, the author’s role with the current paper is as an information compiler, analyzer, and inference drawer.
Response: We thank the reviewer for this insightful observation. We agree that, in this context, the role of the authors is to act as compilers, analysts, and interpreters of the data based on the evidence collected. This perspective reinforces the study’s commitment to rigorous analysis and the production of evidence-based knowledge, grounded in scientific responsibility. The suggested passage was retained in the manuscript, as it accurately reflects the methodological stance adopted by the authors.
Comment 3: The study reflects the COVID period that would have been mentioned (though it is understood from the title).
Response: We thank the reviewer for the observation. We agree that, although the title already implies that the study refers to the COVID-19 period, explicitly stating this in the body of the text is essential to reinforce the temporal context of the investigation. Therefore, the introduction section has been revised to clearly state that the data analyzed pertain to the COVID-19 pandemic period and the corresponding social distancing and isolation measures (IMDIS), as suggested.
Comment 4: Line 115, under number 65610017.1.0000.0008’ does it mean approval number?
Response: Yes
Comment 5: Presentation style, data, and language are the author’s own; inferences presented appear satisfactory. Minor editing aspects may be checked as: Abstract, Line 19, ‘The objective were’- would be “objectives”…. Thus, the full paper may be checked for similar issues.
Response: We thank the reviewer for their valuable suggestions. The manuscript has been carefully revised to incorporate the recommended changes, ensuring greater clarity, coherence, and alignment with the objectives of the study.